# Organically Modified Nanoclay Filled Thin-Film Nanocomposite Membranes for Reverse Osmosis Application

**DOI:** 10.3390/ma12223803

**Published:** 2019-11-19

**Authors:** Syed Javaid Zaidi, Farid Fadhillah, Haleema Saleem, Alaa Hawari, Abdelbaki Benamor

**Affiliations:** 1Center for Advanced Materials, Qatar University, P.O. Box 2713 Doha, Qatar; haleemasaleem@gmail.com; 2Chemical Engineering Department, Al Imam Mohammad Ibn Saud Islamic University, P.O. Box 5701 Riyadh, Saudi Arabia; FFFadhillah@imamu.edu.sa; 3Department of Civil and Architectural Engineering, Qatar University, P.O. Box 2713 Doha, Qatar; a.hawari@qu.edu.qa; 4Gas processing Center, Qatar University, P.O. Box 2713 Doha, Qatar; benamor.abdelbaki@qu.edu.qa

**Keywords:** desalination, polyamide, reverse osmosis, nanoclay, thin-film nanocomposite membrane

## Abstract

This study validates, for the first time, the effectiveness of two nanoclays, that is, cloisite (CS)-15A and montmorillonite (MNT) at the polyamide (PA) active layer in the reverse osmosis (RO) membrane. Cloisite-15A is natural montmorillonite modified with dimethyl dihydrogenated tallow quaternary ammonium salt. Thin-film composite (TFC) membranes were fabricated by the interfacial polymerization (IP) process between the trimesoylchloride (TMC)–n-hexane solution and m-phenylenediamine (MPD)–aqueous solution; the IP process took place on a polysulfone support sheet. The two types of nanoparticles were added in various weight ratios (0.005 wt.%–0.04 wt.%) in the n-hexane solution of TMC. Different characterizations like X-ray diffraction (XRD), contact angle, transmission electron microscopy (TEM), and membrane performance tests were performed to analyse the membrane properties. Both XRD and TEM studies proved that the two nanoclays are successfully anchored at the different sites of the PA layer. CS-15A could accelerate the water flux from 15 to 18.65 L/m^2^·h with NaCl rejection enhancement from 72% to 80%, relative to the control membrane. Conversely, MNT also enhanced the flux from 15 to 40 L/m^2^·h, but NaCl rejection reduced from 70% to 23%. The mechanism of water uptake in nanoclays was also discussed. The results pave the way for a complete future study, in which these phenomena should be studied in great detail.

## 1. Introduction

During the past decade, the scarcity of and requirement for freshwater resources have turned out to be increasingly urgent, brought about by a quick increment in the human population, climate change, and water pollution. Desalination is a practical strategy equipped for giving potable water from seawater. Owing to the tremendous amount of saline water available, as well as the innovative developments for desalination, a great number of commercial-scale seawater desalination plants have been built up globally, and the count is projected to develop significantly [1]. Subsequently, considerable work has been dedicated to advancing the desalination process as well as overpowering operational issues [2]. Reverse osmosis (RO) has been globally recognized as an effective strategy for addressing the scarcity of water resources over the last few decades and is considered to be one of the most common desalination techniques [3]. In this technique, the pressure is utilized for pushing water from a salty side by means of a selective membrane to a permeate side; thereby, the rejection of salt takes place. In order to operate the system, the pressure applied should surpass the osmotic pressure of saltwater and the membrane should just permit molecules of water to go through, and not allow the hydrated salt ions through [4]. Reverse osmosis is considered to be a consistent process even on a commercial-scale; after the initial reverse osmosis plant was set up almost 50 years ago, this strategy ruled the new desalination market almost two decades later. Currently, this technology has a substantial effect on freshwater supplies in various regions [5].

During the mid-1980s, John Cadotte developed an advanced kind of membrane known as a polyamide (PA) thin-film composite membrane (TFCM) [6]. The aforementioned TFCM has turned out to be the most commonly used kind of membrane in the desalination industry owing to its high water permeation and superior desalting properties [7]. The conventional TFC membrane is manufactured by the interfacial polymerization process of trimesoylchloride (TMC) in an organic solution and meta-phenylene diamine (MPD) in aqueous solution, and the reaction happens on a polymer support sheet [6,8,9]. Even though it seems like the polyamide-based TFCMs have matured in reverse osmosis desalination because of their widespread applications, these membranes are still far from optimal. The perfect membrane must have increased water flux per unit of operating pressures, almost entirely NaCl rejections, superior oxidation, as well as fouling resistance. Fabricating superior performance thin-film composite membranes was the objective of numerous researchers, where various components were added, including nano-sized additives, for enhancing the performance of membrane [10,11,12]. The addition of nano-sized additives to the thin-film composite membrane defined a novel membrane category termed as thin-film nanocomposite (TFNC) membranes, which is widely known these days to have contributed an extra dimension to upgrade the membrane efficiency.

With the introduction of nanotechnology, several nanomaterials have attracted tremendous scientific attention for tackling major 21st-century problems. For example, using nanomaterial as a membrane themselves or doping into other composite membranes has shown higher productivities, including pollutants selectivity and retention with high water flux [13]. Reducing the dimension of nanomaterial enhances the specific surface area (SSA) and changes three-dimensional as well as two-dimensional crystallinity, which in turn influences chemical reactivity along with the thermal, mechanical, and chemical properties, which may drive the advancement of more affordable, reliable, efficient, and sustainable novel membrane technologies. This idea of incorporating nanomaterials into the TFC membrane was presented by Jeong et al. [14], in which the team incorporated zeolite A nanocrystals into the TFCM active layer. The aforementioned approach was able to improve the water flux to 16.96 L/m^2^·h from 9.37 L/m^2^·h and retain the NaCl rejection of 93%, operated at 2000 ppm feed salt concentration and 12.4 bar pressure. Scientists have used nanomaterials such as carbon nanotube (CNT) [15], graphene [16], titanium dioxide [17], and silver nanoparticles [18], among others, for fabricating the TFNC membrane. Despite that, naturally occurring nanomaterials such as clay powders are interesting, as well as sustainable sources drawing huge attention among the research community because of their green as well as universal features [19].

Clay minerals are hydrophilic in nature and are hydrated phyllosilicate materials that are typically categorized into three primary classifications of illite (or Mica), smectite, and kaolinite. Out of these three types, the smectite group, especially montmorillonite (MNT), has numerous benefits; for example, low-cost, abundance in nature, non-toxicity, and superior dispersion properties [20]. Montmorillonite is a two-dimensional layered nanomaterial made out of two tetrahedral sheets that are sandwiched between a solitary octahedral sheet. Between the layers, a cation (for example, Na^+^) is positioned. This 2:1 layered smectite clay mineral is shown in Figure 1 [21]. The thickness of the individual platelet is just 1 nm; however, the surface dimensions are normally from 300 nm to greater than 600 nm, leading to a remarkably higher aspect ratio. The naturally occurring MNT is hydrophilic in nature. In a study by Matsuura et al. [22], the team utilized MNT to manufacture clay membranes. The aforementioned membrane demonstrated water flux of 0.35 L/m^2^·h and had the ability to reject the sodium ions until 50%, operated at 5850 ppm feed NaCl concentration and pressure of 30 bar. Most of these nanomaterials were incorporated into the polymer during membrane casting, and the membrane is called a mixed-matrix (MM) membrane. Dong et al. [23] incorporated MNT and layered double hydroxide clay nanosheets (from 100 nm to 200 nm) into the TFC membrane, and the test results illustrated that both the aforementioned clays can upgrade the performance of the membrane as well as its antifouling property. In the work by Kadhom et al. [24], bentonite nanoparticles fabricated by the solvo-thermal technique were added to the TFNC membrane for desalinating the brackish water. An increment in rejection of salt and water flux was attained by incorporating bentonite nanoparticles at a lower concentration.

Cloisite-15A is a natural MNT modified using a quaternary ammonium salt possessing 125 milliequivalents per hundred grams’ clay concentration. This nanoclay is considered to be an additive for rubber as well as plastic for improving different physical properties, for example, barrier, synergistic flame retardant, and reinforcement properties. The organoclay, cloisite, is manufactured from cation di-tallow and the MNT [25]. Tallow is the combination of tetradecyl, hexadecyl, and octadecyl, in which octadecyl is the main component (greater than 60%). The particle size distribution of cloisite-15A (CS-15A) indicates that 90 percent of them are smaller than 13 microns, 50 percent are smaller than 6 microns, and 10 percent are smaller than 2 microns. The chemical structure of the CS-15A modifier is shown in Figure 2. This nanoclay is regarded as a strong potential additive in nanocomposite membranes, owing to the fact that this nanoclay is already organically modified from the natural montmorillonite, which is anticipated to upgrade its compatibility with organic polymers [26]. Sulfonated poly(ether ether ketone) (SPEEK)/CS-15A nanocomposite membranes were manufactured by adding the compatibilizer 2,4,6-triaminopyrimidine and tested for direct methanol fuel cells (DMFC) application [25]. The barrier performance of the aforementioned membrane was remarkably increased to compete with the expensive commercially used Nafion112 membrane.

The utilization of nanoclay CS-15A and MNT in the active layer of the TFC desalination membrane has not been investigated up until now. However, it has been effectively utilized for the MM membrane [28,29], nanocomposite membrane for DMFC application [25], and gas separation membranes [30,31]. There have been numerous research studies carried out for including nanoclays in polyamide membranes, although the objective of these studies was to upgrade the barrier property [32], which is certainly not appropriate for reverse osmosis utilization. Therefore, in this study, our objective is to take a comprehensive examination of the behaviour of two nanoclays (cloisite (CS)-15A and montmorillonite (MNT)) at the polyamide active layer interface to considerably improve the performance of desalination membrane technologies [33]. This study proves that the nanoclay could act as efficient additives for the PA layer to improve water flux as well as the salt rejection capability of a novel TFC-based reverse osmosis membrane.

## 2. Materials and Methods

### 2.1. Materials

The polysulfone used here was obtained from Sigma-Aldrich. Polysulfone is included in the family of thermoplastic polymers and has the formula of (C_6_H_4_C(CH_3_)_2_C_6_H_4_OC_6_H_4_SO_2_C_6_H_4_O)n. The polyvinyl pyrrolidone (PVP, K-90) and *N*-methyl pyrrolidone, used for support preparation, were also from Sigma-Aldrich. The interfacial polymerization reactants, trimesoylchloride (TMC, ≥98.5%) and m-phenylenediamine (MPD, ≥99%), were obtained from Sigma-Aldrich (Sigma-Aldrich, St. Louis, MO, USA) and Fisher Scientific (Thermo Fisher Scientific Houston, Houston, TX, USA), respectively. MNT used in this work was nanoclay monomer^®^ I.34 TCN from Sigma-Aldrich (Qatar, Sigma-Aldrich, St. Louis, MO, USA). The properties of nanoclay MNT are presented in Table 1. Further, the cloisite used in this study was cloisite^®^15A obtained from Neunano. The properties of nanoclay CS-15A are presented in Table 2. The percentage weight loss on ignition of CS-15A was 43%. A millipore deionized (DI) water system (Synergy185, Synergy185, Billerica, MA, USA, 18.2 MΩ cm) was utilized in the current work to make aqueous solutions and to soak as well as flush the membrane specimens in the course of experiments.

### 2.2. Method

Membrane preparation consisted of two main steps, that is, support membrane preparation and polyamide thin layer preparation, as explained in detail by Fadhillah [33]. The structure of the TFC membrane is presented in Figure 3.

#### 2.2.1. Polysulfone Support Membrane Fabrication

Polysulfone (PSf) was selected for preparing the porous, as well as a sponge-like substrate membrane over which a PA TFC was prepared through the in situ interfacial polymerisation process. The aforementioned support was fabricated in the following way: sixteen wt.% of PSf and three wt.% of polyvinylpyrrolidone (PVP) were dissolved in *N*-methyl-pyrrolidone (NMP). This solution underwent stirring for a few hours until both PSf and PVP were totally dissolved. This solution was then distributed with knife-edge on top of a non-woven polyester fabric fixed to a glass plate. Subsequently, the aforementioned glass plate was submerged instantaneously into a DI water bath with a temperature of 20 °C. A few minutes later, the non-woven fabric support, along with the PSf membrane, was detached from the glass plate. This membrane was later flushed properly with DI water and then reserved in a refrigerator at a temperature of 5 °C prior to usage.

#### 2.2.2. Preparation of Thin Polyamide Layer

The support membrane (polysulfone + nonwoven polyester fabric) fixed to a glass plate was submerged for about 30 min in an aqueous solution consisting of 2 wt.% of m-phenylenediamine. The excess m-phenylenediamine solution was blotted from the surface of the membrane before the membrane was immersed into a solution of 0.1 wt.% trimesoyl chloride in n-hexane. By holding the membrane vertically, the trimesoyl chloride solution was drained off, and later the membrane was held at room temperature. At this course, the in situ interfacial polymerisation occurred. The resultant membrane was then flushed with a 0.2 wt.% sodium carbonate (Na_2_CO_3_) aqueous solution prior to being stored in DI water. This membrane later underwent heating in an air circulating oven at temperature 105 °C for about 90 s to facilitate the polymerization process. The resultant membrane was once more flushed with 0.2 wt.% sodium carbonate aqueous solution.

The inclusion of MNT and CS 15A into the PA layer was done by the dispersion of a definite amount of the aforementioned nanoclays (from 0.005 wt.% to 0.04 wt.%) in trimesoyl chloride–hexane solution. This mix was later ultrasonicated for about 1 h before immersing on the polysulfone membrane. The remaining procedure was similar to the interfacial polymerization process with no nanoclays.

### 2.3. Characterization of Membranes

#### 2.3.1. X-Ray Diffraction (XRD)

The crystalline structures of MNT and CS-15A TFC membranes were assessed by powder X-ray diffraction (PXRD). This was undertaken using an X-ray diffractometer D8 of Bruker AXS with Cu Kα radiation. The aforementioned characterization was employed for examining the degree of intercalation of the MNT and CS-15A in polyamide film.

#### 2.3.2. Transmission Electron Microscopy (TEM)

FEI Tecnai 12 (T12) transmission electron microscopy was employed for analyzing the morphology of cloisite 15A, as well as the montmorillonite-embedded TFC membrane. Prior to transmission electron microscopy characterization, a smaller piece of membrane specimen cross-section was loaded in epoxy. Later, the aforementioned loaded specimen was trimmed, accompanied by slicing, employing a trimmer and microtome to obtain almost 80 nm thick samples. The aforementioned thin samples were later examined utilizing the transmission electron microscopy.

#### 2.3.3. Contact Angle Measurement

The hydrophilicity of the membrane was estimated based on the contact angle between the surface of the membrane and a pure water drop. Utilizing DM-501 goniometer (Kyowa Interface Science Co., Saitama, Japan), the sessile drop contact angle analysis was performed to examine the hydrophilicity of the membrane. Deionized water of about 2 L was dispensed and then dropped on to the membrane surface, and right as well as left contact angles were assessed instantly.

### 2.4. Membrane Performance Tests

Employing the CF042 (Sterlitech Corp., Kent, WA, USA) cross-flow permeation cell, the permeation tests were carried out. Water flux and NaCl rejection were assessed by a cross-flow filtration system; a diagrammatic illustration of the testing system is shown in Figure 4. The permeation cell analyzed an active membrane surface area of almost 42 cm^2^. The test was performed at a pressure of 40 bars, pH of 6, and at 2000 ppm feed NaCl concentration. The membrane was tested for about 40 h, whereas the temperature of feed was kept at 25 °C with the aid of constant temperature circulating baths (Polystat, Cole-Parmer Instruments Co., Vernon Hills, IL, USA). The enhanced controller, as well as the temperature sensor in Polystat, confirm accurate and repeatable analyses. Prior to the permeation test using saline water, the membrane was conditioned using distilled water for about 2 h at a pressure of 1000 psi. Later, the feed was changed to saline water, and then the permeate flux, as well as conductivity for both permeate and feed solutions, were examined at particular intervals of almost every 1 h.

## 3. Results and Discussion

### 3.1. Characterization Analyses

#### 3.1.1. X-Ray Diffraction

Cloisite-15A is a natural montmorillonite modified using quaternary ammonium salt dimethyl dihydrogenated tallow (DMDT). This tallow contains 5% of C14, 30% of C16, and 65% of C18. Cloisite-15A is increasingly hydrophobic as compared with natural montmorillonite. The montmorillonite is categorized as 2:1 clay and is included in the smectite family. In this nanoclay, two tetrahedral layers are sandwiched on an octahedral layer. Characteristic gallery inter-spacing of montmorillonite is almost 1 nm to 2 nm and is extended because of the modification process; for example, 3.15 nm gallery spacing for cloisite-15A. In the course of TFC membrane fabrication, the nanoclays MNT and cloisite-15A are dispersed in the organic phase, owing to the fact that the PA film builds up towards the organic phase.

XRD patterns of MNT and CS 15A modified PA membranes are shown in Figure 5 and Figure 6, respectively. Tracking the position as well as the intensity of XRD patterns can be a beneficial means in learning intercalated and exfoliated nanostructures. Pristine MNT displays a major crystalline peak at 6.85° [35], but, in our specimen, the aforementioned peak is moved towards a smaller degree, that is, at 4.9° with greater d-spacing, as shown in Figure 5. It is recommended that the nanoclay montmorillonite is intercalated [36]. In the course of ultrasonication for 1 h, the MPD diffuses within the gallery spacing of MNT and enlarges it. Subsequent to this enlargement, the next monomer, that is, TMC could straightforwardly diffuse within the gallery and undergo reaction with MPD for developing PA. The nanoclay cloisite-15A has two distinguishing peaks appearing at 3° as well as 7° 2θ [37]. Figure 6 shows a peak appearing at 5°, which is supposed to be a peak originally positioned at 7°. The next peak, that is, at 3°, has been moved to lower than 2° 2θ, which was not scanned employing this XRD. The aforementioned is again evidence of CS-15 intercalation in a PA membrane.

#### 3.1.2. Transmission Electron Microscopy

TEM characterization technique was employed to observe the cross section of the membrane and identify the presence of montmorillonite and cloisite-15A nanoparticles within the polyamide thin-film composite membrane. Figure 7 shows transmission electron microscopy images of a cross section of MNT (a) and CS-15A (b) embedded TFC PA membranes. According to Figure 7, it is clearly visible that both the MNT and CS-15A were effectively incorporated into the polyamide active layer. The extremely porous polysulfone layer is coated with a polyamide layer, of 250 nm average thickness. Also, it could be observed that the surface of the membrane is relatively rough, which might be because of the agglomeration of MNT and CS-15(A).

In spite of the fact that nanoclays are intercalated, as observed from X-ray diffraction results, it was noticed from transmission electron microscopy images that montmorillonite is not properly dispersed. This is because agglomeration happens and gives rise to a relatively large chunk (Figure 7a). As MNT is a hydrophilic material, which is less soluble in organic TMC solution, it might cause the agglomeration of MNT in the PA layer. On the contrary, because CS-15A is hydrophobic as compared with MNT, it shows better dispersion in the TMC solution, so that the uniform deposition of CS-15A occurs into the PA layer, as expected (Figure 7b).

### 3.2. Performance Analysis

#### 3.2.1. Nanoparticle Concentration and Performance of the Membrane

From Figure 8a, it was seen that the incorporation of lower concentration MNT into PA improves the membrane performance. It was observed that, as the concentration of montmorillonite is increased, the water flux enhanced from 15 L/m^2^·h to around 40 L/m^2^·h. It seems that the increment in water flux is possibly because of additional channels resulting from MNT agglomeration. Further, these channels evidently offer a pathway for NaCl to move across the membrane, and this leads to a remarkable reduction in the rejection of NaCl from 70% to just 23%.

However, the permeation performance of the CS-15A-filled thin-film composite membrane is a little different from that of the MNT-filled TFC membrane (Figure 8b). It was found that CS-15A follows a similar trend to other kinds of nanoclays, which typically display better barrier properties of the polymer. It could be seen from Figure 8b that the water flux reduces as cloisite-15A concentration is increased at the beginning. The aforementioned trend demonstrated that the cloisite-15A dispersed as well as intercalated relatively well, thereby providing a tortuosity effect on the membrane, which ultimately increases the barrier properties. On the other hand, the barrier property rapidly reduced at a CS-15A concentration of 0.02 wt.%, illustrated by remarkable water flux improvement, which might develop from cloisite-15A agglomeration. Unpredictably, the flux improvement is coupled with the rejection increment because of the enhancement in surface charge brought about by CS15A. It was noted that the specific orientation of cloisite-15A platelets was lost at cloisite-15A loading greater than 0.2 wt.%. Nevertheless, as a general rule, the orientation of the platelets is still capable of, at the very least, retaining the water flux as well as NaCl rejection greater than those of pure TFC polyamide membranes.

Also, a separate study was conducted by heating the membrane in an air circulating oven at a temperature of about 105 °C for 90 s in order to facilitate the in situ interfacial polymerization process for the heat treatment. The performance testing was conducted as follows. All membranes were compacted under pure feed water at 1000 psi for 2 h. Then, the feed was changed to 3.5 wt.% NaCl (food grade table salt) solution. The operating pressure was decreased to 800 psi. TFC-MNT-0-(105,90) indicates a TFC membrane fabricated using the MNT substrate membrane, with 1 wt.% of MNT particles and heated at 105 °C for 90 s. Here, no nano-clay is added to the TMC–hexane solution. Similarly, TFC-CS-0-(105,90) is the TFC membrane fabricated using the CS-15A substrate membrane, with 1 wt.% of CS-15A particles and heated at 105 °C for 90 s. The results confirmed that the salt rejection for TFC-MNT-0-(105,90) and TFC-CS-0-(105,90) increased to 98.7% and 78.9%, respectively; however, the fluxes were considerably lower than the best commercial membranes

#### 3.2.2. Membrane Hydrophilicity and Performance

Figure 9 displays the relation between the hydrophilicity of the membrane and its performance. Both MNT and cloisite-15A seem to not improve the membrane hydrophilicity. The two nanoclay-filled membranes showed a contact angle value of approximately 60°, with MNT-PA showing a higher contact angle value than cloisite-15A-PA. The contact angle is dependent not only on the surface hydrophilicity (or surface hydrophobicity), but also on the roughness. The contact angle value of a membrane of greater surface roughness is typically greater relative to another membrane of lesser surface roughness, even though the two membranes have the same hydrophilic nature [38]. Similarly, contact angle values of membranes of higher surface roughness are comparable, though the membranes are, to some extent, diverse in hydrophilic character. From our perspective, the montmorillonite-filled TFC membrane does show greater hydrophilicity relative to the cloisite-15A-filled TFC membrane; however, as the membrane is very rough (Figure 7), the impact of roughness oppresses the effect of MNT.

### 3.3. Mechanism of Water Uptake

The nanoclays possess the water uptake ability within their structural layers, once they are exposed to water. In the water treatment membrane application, the phenomenon of water uptake can affect the process in two ways. The first one is at the time of TFNC membrane fabrication, where the nanoclay might adsorb the IP reactants, particularly if the nanoclay is included in the MPD solution. The next one is in the course of the membrane filtration process. During membrane fabrication, the nanoparticles are generally loaded to either an aqueous solution of MPD or an organic solution of TMC. The nanoparticles added to the solutions will have the ability to adsorb the solution with the IP reactant in their pores, which give the preparation process two scenarios depending on the filling method. In the case when the nanoparticles are added into the MPD solution, the nanoparticles will adsorb the MPD along with the water, within the pores. At the time when the excess MPD solution is separated from the surface of the support layer, these nanoparticles will still retain certain solution within the pores. Later, when the TMC solution is added, the IP reactant penetrates into the nanoparticle pores, and then the IP process might happen and develop a thin layer of polyamide on the walls. During the second filling technique, that is, when the nanoparticles are added into the TMC solution, the nanoparticles adsorb the TMC along with the organic solution, within the pores of nanoparticles. By the addition of TMC solution to the MPD-support layer, the IP process occurs, although at a low rate, owing to the fact that the MPD amount is not overwhelming the TMC. This is because the excess MPD solution is separated from the support layer top prior to the addition of the TMC solution. Thus, the addition of nanoparticles to the TMC solution can develop lesser micropore thinning as well as blocking, relative to the nanoparticle addition to the MPD solution. This can be one reason for why scientists usually obtain superior results by the addition of nanoparticles to the TMC solution, relative to the MPD solution, with regard to other factors [24]. The additional reactants can be the reason for enhanced hydrophilicity, owing to the fact that the acyl chlorides (in TMC) might hydrolyze to generate COOH, or the amine groups (in MPD) can be oxidized to develop amine oxide [9].

Smectite shows the capability to adsorb enormous amounts of water, thereby developing a water-tight barrier. MNT is a soft smectite layered silicate with plate-like particles, and the quaternary ammonium salt modified form of MNT, known as cloisite, demonstrate a hydrophobic nature owing to the increased interlayer space existing because of the existence of long alkyl chains [39]. The aforementioned makes cloisite compatible with a polymer matrix. Owing to the existence of cations in the smectite interlayers of clay, the water adsorbed could be in the hydration layer of ions, thereby developing the structured water layers within the intermediate layers. The aforementioned hydrated ions could be replaceable and could lead to structures possessing superior hydrophilicity, whereas the uncharged spacing between the clay structural layers demonstrates lesser water affinity [40]. The uptake of water is influenced by several different factors, inclusive of the charge density, cation species, smectite type, and location [41]. A few studies have addressed the swelling ability of nanoclay MNT. The water molecules can develop swelling in MNT, which occurs because of the complicated MNT–water interactions. The adsorption of water molecules and swelling of MNT contribute to hydrated states as well as hysteresis. The counter-ion migration, initially bound to MNT surface to the central interlayer plane, results in the MNT swelling. Hence, charge locus in MNT has a powerful impact on the swelling dynamics [42]. The MNT platelets could be negatively charged while (1) the tetrahedral replacement of Si by Al in both tetrahedral sheets occurs, or (2) the octahedral replacement of Al by Mg in the central octahedral sheet occurs. In these cases, the negative charge developed is compensated by the interlayer ions. The interlayer cation hydration develops swelling [43]. Discrete smectite particles have multiple layers of “fundamental particles” of 1 nm thickness, which are usually stacked over the other, as shown in Figure 10. For Na^+^ and Li^+^ at lower ionic strength, the particles could be totally disarticulated into single one nanometer thick particles. In the case of other cations, seven layers could be stacked over one another in structures alluded to as tactoids, as represented in Figure 10. 

Molecular dynamic simulation (MDS) is a beneficial method to understand the atomic level configuration, and is suitable to examine the MNT framework inclusive of swelling as well as hydration of interlayer cations. MDS was carried out by Faheem et al. [45] for Ca^2+^-, K^+^-, and Na^+^-MNTs, with a changing level of water content. The simulations illustrate that the cation K^+^ showed a powerful interaction with dehydrated MNT sheets; but for hydrated MNT sheets, the cation Ca^+^ interacted intensely. Consequently, the layer spacing of simulated Ca^2+^-, K^+^-, and Na^+^-MNTs was obvious [45].

## 4. Conclusions

Because nanomaterials are receiving increased attention in the membranes field these days, this work studied the impact of two different nanoclays (that is, cloisite 15A and montmorillonite) on the performance of a TFNC membrane. The two distinct nanoclays were introduced in a polyamide-based RO membrane for the first time, with the aim of improving the performance of the membrane. TFC membranes were fabricated using the in situ interfacial polymerization process between an n-hexane solution of TMC and an aqueous MPD solution. The two types of nanoparticles were incorporated in various weight ratios, from 0.005 wt.% to 0.04 wt.% in the TMC–hexane solution. XRD, TEM, contact angle, and membrane performance tests were carried out to characterize the properties of the membrane. The results showed that CS-15A could enhance the water flux from 15 L/m^2^·h to 18.65 L/m^2^·h by the addition of just 0.02 wt.% nanoclay, and also increase the salt rejection from 72% to 80%. The aforementioned improvement is because of the increased interaction between the organic solvent used in PA synthesis and CS-15A. Further, it was observed that MNT nanoclays also improve the flux immensely, although they result in a substantial reduction in the rejection of salt, demonstrating a certain extent of agglomeration occurring in the membrane. The aforementioned occurred because of the poor interaction occurring between the organic solvent utilized in PA synthesis and MNT. This study opens up the feasibility of utilizing additional organically modified nanoclays, which possibly contribute to improved interaction with the organic solvent used in the polyamide synthesis; consequently, substantial improvement in the membrane performance can be obtained.

## Figures and Tables

**Figure 1 materials-12-03803-f001:**
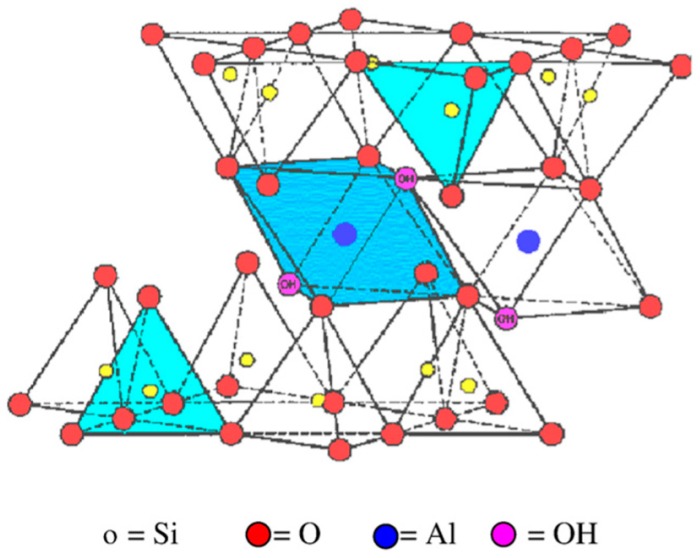
Structure of montmorillonite (MNT). Reproduced with permission from the authors of [21].

**Figure 2 materials-12-03803-f002:**
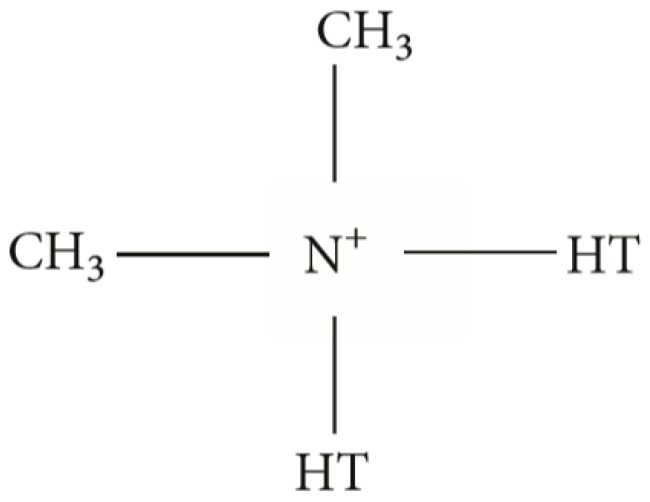
Molecular structure of Cloisite-15A modifier. Reproduced with permission from the authors of [27].

**Figure 3 materials-12-03803-f003:**
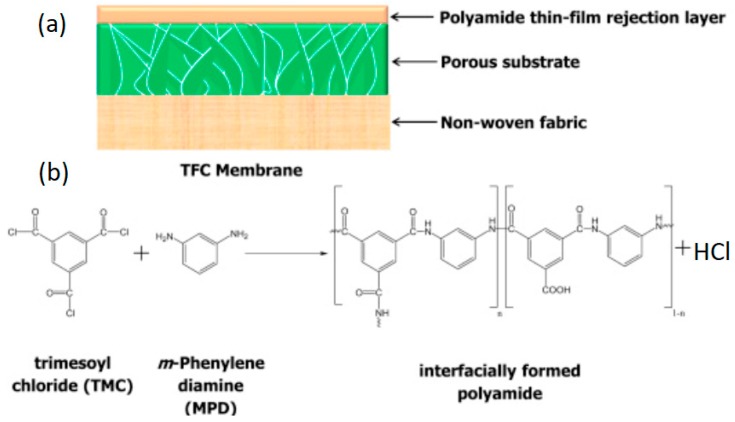
Structure and preparation of the thin-film composite (TFC) polyamide (PA) membrane: (**a**) structure of a typical membrane; (**b**) interfacial polymerization reaction between MPD and TMC. Reproduced with permission from the authors of [34].

**Figure 4 materials-12-03803-f004:**
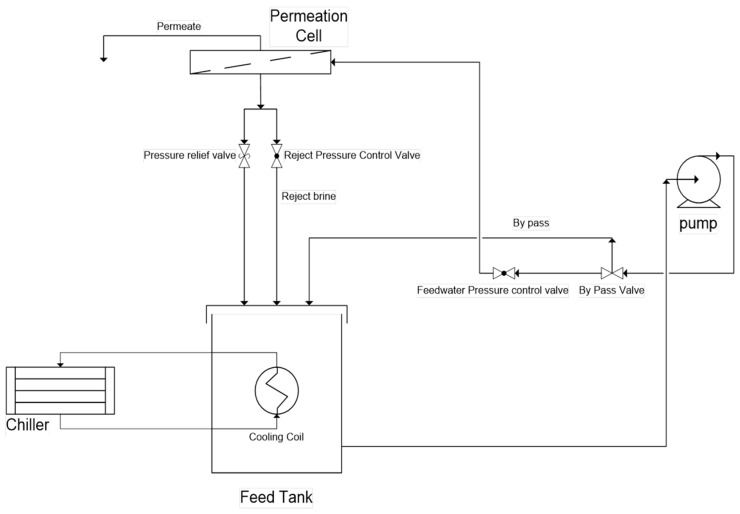
A diagrammatic representation of the cross-flow filtration system.

**Figure 5 materials-12-03803-f005:**
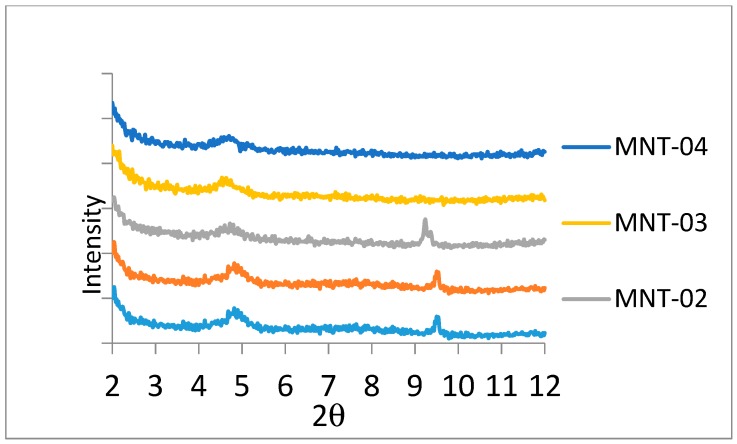
X-ray diffraction characteristic peaks for an MNT-filled PA membrane. The inclusion of MNT into the PA layer was done by the dispersion of a definite amount of nanoclays (0.005 wt.%, 0.01 wt.%, 0.02 wt.%, 0.03 wt.%, and 0.04 wt.%) in TMC-hexane solution. MNT-005, MNT-01, MNT-02, MNT-03, and MNT-04 refer to MNT incorporation of 0.005 wt.%, 0.01 wt.%, 0.02 wt.%, 0.03 wt.%, and 0.05 wt.%, respectively.

**Figure 6 materials-12-03803-f006:**
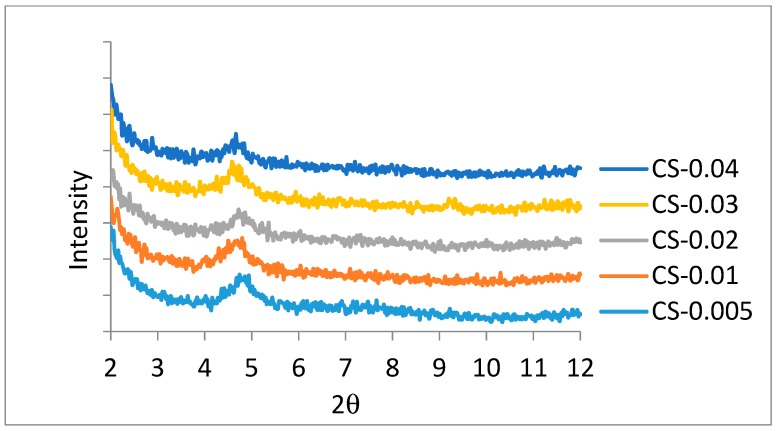
X-ray diffraction characteristic peaks for a cloisite-15A (CS-15A)-filled PA membrane. The inclusion of CS 15A into the PA layer was done by the dispersion of a definite amount of the nanoclays (0.005 wt.%, 0.01 wt.%, 0.02 wt.%, 0.03 wt.%, and 0.04 wt.%) in TMC–hexane solution. CS-0.005, CS-0.01, CS-0.02, CS-0.03, and CS-0.04 refer to CS-15A incorporation of 0.005 wt.%, 0.01 wt.%, 0.02 wt.%, 0.03 wt.%, and 0.05 wt.%, respectively.

**Figure 7 materials-12-03803-f007:**
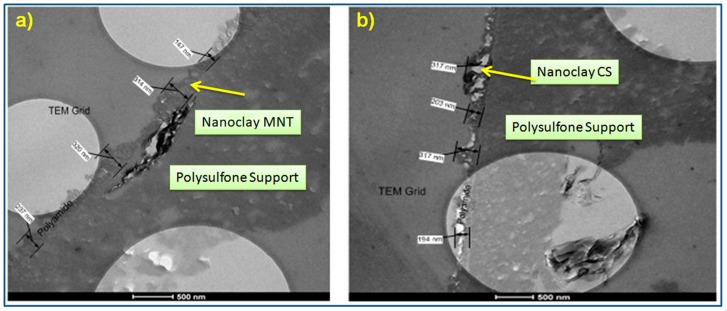
Cross-section transmission electron microscopy image of MNT (**a**) and CS-15A (**b**) modified PA membranes (0.04 wt.% MNT).

**Figure 8 materials-12-03803-f008:**
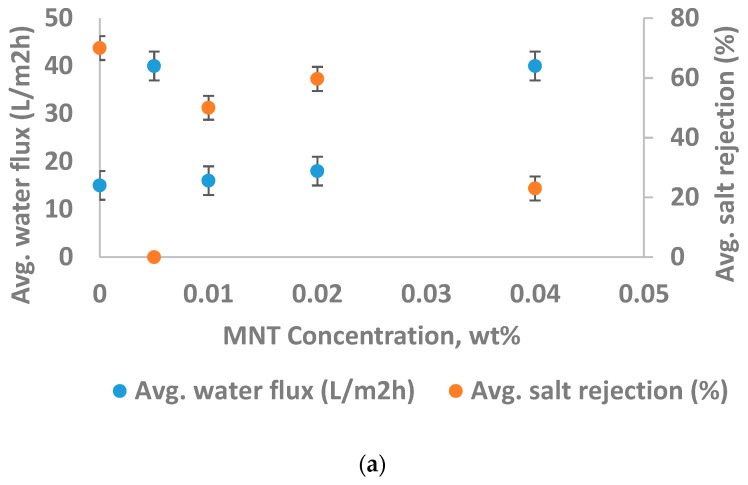
Impact of (**a**) montmorillonite (MNT) concentration and (**b**) cloisite-15A (CS-15A) loading on the performance of the membrane. The test was performed at a pressure of 40 bars, 6 pH, 2000 ppm feed NaCl concentration, and feed temperature of 25 °C.

**Figure 9 materials-12-03803-f009:**
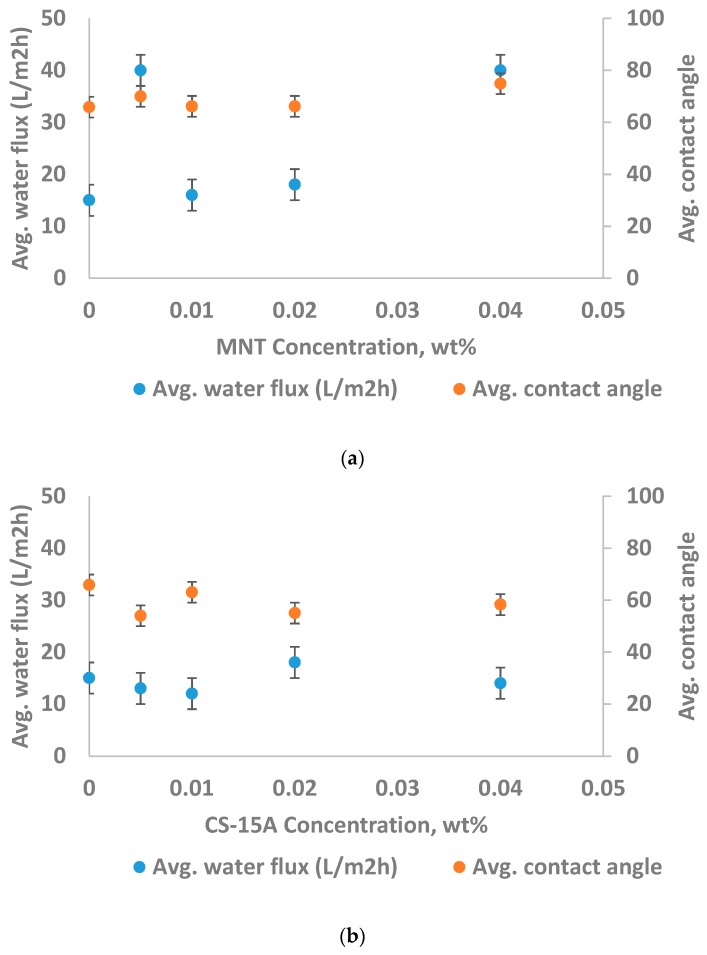
Impact of (**a**) montmorillonite (MNT) concentration and (**b**) cloisite-15A (CS-15A) loading on the performance of a membrane, as well as membrane hydrophilicity (the test was conducted at a feed temperature of 25 °C, a pressure of 40 bars, pH of 6, and a feed NaCl concentration of 2000 ppm).

**Figure 10 materials-12-03803-f010:**
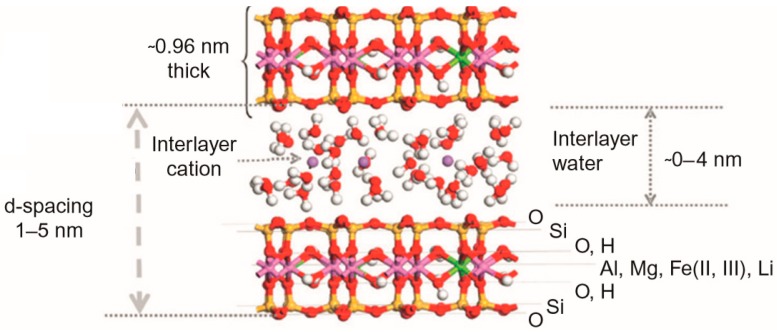
Diagrammatic representation of a three-layer hydrate. Reproduced with permission from the authors of [44].

**Table 1 materials-12-03803-t001:** Montmorillonite (MNT) physical properties. XRD, X-ray diffraction; CS-15A, cloisite-15A.

Physical Properties	MNT
Density in g/cm^3^	2.80
Particle (clay platelet) size in m	≤2 (10%); ≤10 (50%); ≤25 (90%).
XRD d-Spacing in nm	1.55
Moisture content in %	≤9.0%

**Table 2 materials-12-03803-t002:** CS-15A physical properties.

Physical Properties	CS-15A
Density in g/cm^3^	1.66
Particle (clay platelet) size in m	≤2 (10%); ≤6 (50%); ≤13 (90%)
XRD d-Spacing in nm	3.15
Moisture content in %	≤2.0

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
