# Peer review of "Organically Modified Nanoclay Filled Thin-Film Nanocomposite Membranes for Reverse Osmosis Application"

_materials, 2019, doi:10.3390/ma12223803_

Round 1
Reviewer 1 Report
After reading this manuscript, I feel it can be published after some minor corrections.
The discussion on Mechanism of Water Uptake is based on references and therefore can be made less wordy. Some experimental conditions do not need to appear in the conclusion. The format of references have to be in line with the requirements. P9 L152 m-phenyldiamine or m-phenylenediamine ?? There is no indication of permission for the use of the copied figures from the literature (Fig 1, 2, 10).
Author Response
We are very much thankful to the external reviewer for the important and valuable comments and suggestions to improve the quality of the manuscript. After considering the reviewers’ comments, we have thoroughly revised the manuscript. Our item-wise responses to Reviewers’ Comments (responses have been shown in green) are given below. In the revised manuscript, the revised text is shown in red color. All page numbers refer to the revised file with tracked changes.
Reviewer 1
Comment: The discussion on Mechanism of Water Uptake is based on references and therefore can be made less wordy.
Authors response: Thank you very much for the kind advise. We have considered the valuable comments and shortened section 3.3. Kindly check the text highlighted in red colour in section 3.3. We trust this is fine now.
Comment: Some experimental conditions do not need to appear in the conclusion.
Authors response: Many thanks for the suggestion. After considering this, we have removed the experimental conditions mentioned in the conclusion section. Hope this is OK now.
Comment: The format of references have to be in line with the requirements.
Authors response: Well noted. Thank you very much. We have now thoroughly revised the reference section to match the requirements of MDPI referencing style. Kindly check the text highlighted in red colour in section 6. We trust this is fine now.
Comment: P9 L152 m-phenyldiamine or m-phenylenediamine ??
Authors' response: Thank you very much for pointing this. We apologize for this mistake. This has now been corrected in page 8. We trust this is fine now.
Comment: There is no indication of permission for the use of the copied figures from the literature (Fig 1, 2, 10).
Authors response: Please note, we have already obtained permission for the use of the copied figures from the literature (Fig 1, 2, 10). We trust this is fine now.
Reviewer 2 Report
The manuscript describes TFC membranes that incorporate nano clay.
A few things should be addressed before publication:
Lines 112, 171: Do the authors have the formal permission to re-use these figures? This should be stated in the caption "reproduced with permission".
Fig. 3: PSL - should this be PSf?
Fig 5, 6 : no need for green background
Fig 5, 6 caption should include definitions of what is presented in the figure legend. What is the meaning of 04, 03 etc.
Fig 8a: The membrane synthesis of sample 005 wt% looks like it failed. Why wasn't this resynthesized? Did you only make 1 synthetic attempt? These results should be repeated so that the results can be an average of at least 2 membranes. Thus you should include error bars on the graphs, and plot average values.
Fig 9a: Same as above, the sample 005 is far out from the trend. I highly recommend to resynthesize and check again. Something probably went wrong in the synthesis.
recommendation: major revision
Author Response
We are very much thankful to the external reviewer for the important and valuable comments and suggestions to improve the quality of the manuscript. After considering the reviewers’ comments, we have thoroughly revised the manuscript. Our item-wise responses to Reviewers’ Comments (responses have been shown in green) are given below. In the revised manuscript, the revised text is shown in red color. All page numbers refer to the revised file with tracked changes.
-Reviewer-2
Comments: Lines 112, 171: Do the authors have the formal permission to re-use these figures? This should be stated in the caption "reproduced with permission".
Authors' response: Thank you very much for pointing this. Please note, we have already obtained permission for the use of the copied figures from the literature. We trust this is fine now.
Comments: Fig. 3: PSL - should this be PSf?
Authors' response: Thank you very much for pointing this. We apologize for this mistake. This has now been corrected and changed to PSF in page 9. We trust this is fine now.
Comments: Fig 5, 6: no need for green background
Authors' response: Thank you very much for the suggestion. However, please note, for Fig 5 and 6, the background was not in green, and it was in white color only. We trust this is fine.
Comments: Fig 5, 6 caption should include definitions of what is presented in the figure legend. What is the meaning of 04, 03 etc.
Authors response: Many thanks for the kind advise. Considering this, we have now included the definitions of all the samples mentioned in Fig 5 and 6 captions. Please check the text highlighted in red colour. Hope this is OK.
Comments: Fig 8a: The membrane synthesis of sample 005 wt% looks like it failed. Why wasn't this resynthesized? Did you only make 1 synthetic attempt? These results should be repeated so that the results can be an average of at least 2 membranes. Thus you should include error bars on the graphs, and plot average values. Fig 9a: Same as above, the sample 005 is far out from the trend. I highly recommend to resynthesize and check again. Something probably went wrong in the synthesis.
Authors response: These test results are the average values of the two-three samples. We had analyzed the results and synthesized the samples to repeat the analysis, however, we were getting the same values. We understand that that particular sample is out of the normal trend. Also, we did not try to remove this particular point in the figure and presented the results as obtained.
The figures 8 and 9 have been plotted again with the error bar on the graph, as suggested by the reviewer.
We have incorporated all the reviewers’ comments and explained them and hope that the manuscript will be accepted now.
Round 2
Reviewer 2 Report
The authors have sufficiently corrected the manuscript